# Friendly-rivalry solution to the iterated $n$-person public-goods game

Yohsuke Murase[ID][1], Seung Ki Baek[ID][2]*

**1** RIKEN Center for Computational Science, Kobe, Japan, **2** Department of Physics, Pukyong National University, Busan, Korea

* seungki@pknu.ac.kr

**Data Availability Statement:** All the data used in this paper are reproducible from the codes available at https://github.com/yohm/sim_CAPRI_nplayers.

**Funding:** Y.M. acknowledges support from Japan Society for the Promotion of Science (www.jsps.

## Abstract

Repeated interaction promotes cooperation among rational individuals under the shadow of future, but it is hard to maintain cooperation when a large number of error-prone individuals are involved. One way to construct a cooperative Nash equilibrium is to find a 'friendly-rivalry' strategy, which aims at full cooperation but never allows the co-players to be better off. Recently it has been shown that for the iterated Prisoner's Dilemma in the presence of error, a friendly rival can be designed with the following five rules: Cooperate if everyone did, accept punishment for your own mistake, punish defection, recover cooperation if you find a chance, and defect in all the other circumstances. In this work, we construct such a friendly-rivalry strategy for the iterated $n$-person public-goods game by generalizing those five rules. The resulting strategy makes a decision with referring to the previous $m = 2n - 1$ rounds. A friendly-rivalry strategy for $n = 2$ inherently has evolutionary robustness in the sense that no mutant strategy has higher fixation probability in this population than that of a neutral mutant. Our evolutionary simulation indeed shows excellent performance of the proposed strategy in a broad range of environmental conditions when $n = 2$ and 3.

## Author summary

How to maintain cooperation among a number of self-interested individuals is a difficult problem, especially if they can sometimes commit error. In this work, we propose a strategy for the iterated $n$-person public-goods game based on the following five rules: Cooperate if everyone did, accept punishment for your own mistake, punish others' defection, recover cooperation if you find a chance, and defect in all the other circumstances. These rules are not far from actual human behavior, and the resulting strategy guarantees three advantages: First, if everyone uses it, full cooperation is recovered even if error occurs with small probability. Second, the player of this strategy always never obtains a lower long-term payoff than any of the co-players. Third, if the co-players are unconditional cooperators, it obtains a strictly higher long-term payoff than theirs. Therefore, if everyone uses this strategy, no one has a reason to change it. Furthermore, our simulation shows that this strategy will become highly abundant over long time scales due to its robustness against the invasion of other strategies. In this sense, the repeated social dilemma is solved for an arbitrary number of players.

go.jp) (JSPS KAKENHI; Grant no. 18H03621). S.K. B. acknowledges support by Basic Science Research Program through the National Research Foundation of Korea funded by the Ministry of Education (www.moe.go.kr) (NRF-2020R1I1A2071670). The funders had no role in study design, data collection and analysis, decision to publish, or preparation of the manuscript.

**Competing interests:** The authors have declared that no competing interests exist.

## Introduction

The success of *Homo sapiens* can be attributed to its ability to organize collective action toward a common goal among a group of genetically unrelated individuals [1], and this ability is becoming more and more important as the world is getting close to each other. Researchers have identified several mechanisms to promote cooperation in terms of evolutionary game theory [2, 3]. For example, the folk theorem holds that repeated interaction can establish cooperation through reciprocal strategies, and this mechanism is called direct reciprocity [4]. Yet, how to resolve a conflict between individual and collective interests is a hard problem, especially when a large number of players are involved and they are prone to error [5–7], because an individual player has very limited control over co-players.

In this respect, the discovery of the zero-determinant (ZD) strategies in the iterated prisoner's dilemma (PD) has been deemed counter-intuitive because a ZD-strategic player can unilaterally fix the co-player's long-term payoff or enforce a linear relationship between their long-term payoffs [8]. For instance, one can design an *extortionate* ZD strategy, with which the player's long-term payoff will increase by $\chi \geq 1$ whenever the co-player's does by one unit payoff. Another counter-intuitive aspect of the ZD strategy is that it is a memory-one strategy referring only to the previous round, so that such a simple strategy can perfectly constrain the co-player's long-term payoff regardless of the co-player's strategic complexity. Of course, the excellent performance in a one-to-one match does not necessarily mean evolutionary success: It is difficult for an extortionate strategy to proliferate in a population because, as its fraction increases, two extortionate players are more likely to meet and keep defecting against each other [9–12]. For this reason, especially in a large population, selection tends to favor a *generous* ZD strategy whose long-term payoff does not exceed the co-player's [11]. A generous ZD strategy does not aim at winning a match, but it is efficient by forming mutual cooperation when they meet each other.

The important point in this line of thought is that a player's strategy can unilaterally impose constraints on the co-player's long-term payoff, so that we can now characterize strategies according to the constraints that they impose. One such meaningful characterization scheme is to ask if a strategy works as a 'partner' or as a 'rival' [13, 14]: By 'partner', we mean that the strategy seeks for mutual cooperation, but that it will make the co-player's payoff less than its own if the co-player defects from it. It has also been called 'good' [15, 16], and the generous ZD strategies can be understood as an intersection between the ZD and partner strategies [11]. On the other hand, a rival strategy always makes its long-term payoff higher than or equal to the co-player's, so it has been called 'unbeatable' [17], 'competitive' [13], or 'defensible' [18, 19]. A trivial example of a rival strategy is unconditional defection (AllD), and an extortionate ZD strategy also falls into this class. Most of well-known strategies in the iterated PD game are classified either as a partner or as a rival [14]. However, which class is more favored by selection depends on environmental conditions such as the population size and the benefit-to-cost ratio of cooperation: If the population is small and cooperation is costly, it is better off to play a rival strategy than to play a partner strategy, and vice versa [11, 14, 20]. In the iterated PD game, if a single strategy acts as a partner and a rival simultaneously, it has important implications in evolutionary dynamics because it possesses *evolutionary robustness* regardless of the environmental conditions [21], in the sense that no mutant strategy can invade a population of this strategy with greater fixation probability than that of neutral drift [11, 20, 22]. To indicate the partner-rival duality, such a strategy will be called a 'friendly rival' [21]. Tit-for-tat (TFT), a special ZD strategy having $\chi = 1$, is a friendly rival in an error-free environment [14], but a friendly rival generally requires a far more complicated structure in the presence of error. So far, the existence of friendly-

rivalry strategies has been reported by a brute-force enumeration method in the iterated PD game [18, 21, 23] and the three-person public-goods (PG) game [19]. However, it is not straightforward to extend these findings to the general *n*-person PG game. For example, a naive extension of a solution in the iterated PD game fails to solve the three-person PG game because the third player cannot tell if one of the co-players is correcting the other's error with good intent or just carrying out a malicious attack [19]. To resolve the ambiguity, a strategic decision must be based on more information of the past interactions: In fact, if a player refers to the previous *m* rounds to choose an action in the *n*-person PG game, we can show that *m* must be greater than or equal to *n* as a necessary condition to be a friendly rival [19]. Unfortunately, the existing brute-force approach then becomes simply unfeasible because the number of possible strategies expands super-exponentially as $2^{2^{mn}}$. For example, in the three-person game ($n = 3$), it means that we have to enumerate $2^{512} \sim 10^{154}$ possibilities to find an answer. Although the symmetry among co-players reduces this number down to $2^{288}$ $\sim 10^{86}$, it is still comparable to the estimated number of protons in the universe.

In this work, by using an alternative method to generalize behavioral patterns of a friendly rival for the iterated PD game [21], we construct a friendly-rivalry strategy for the *n*-person PG game. This approach makes use of the fact that it greatly lessens the computational burden if we only check whether a given strategy qualifies as a friendly rival. The required memory length of our strategy is $m = 2n - 1$, which satisfies the necessary condition $m \geq n$ as shown in Fig 1. We will also numerically confirm that it shows excellent performance in evolutionary dynamics. In this way, this work modifies and generalizes our previous finding of $n = 2$ and $m = 3$, i.e., a memory-three friendly-rivalry strategy for the iterated PD game [21].

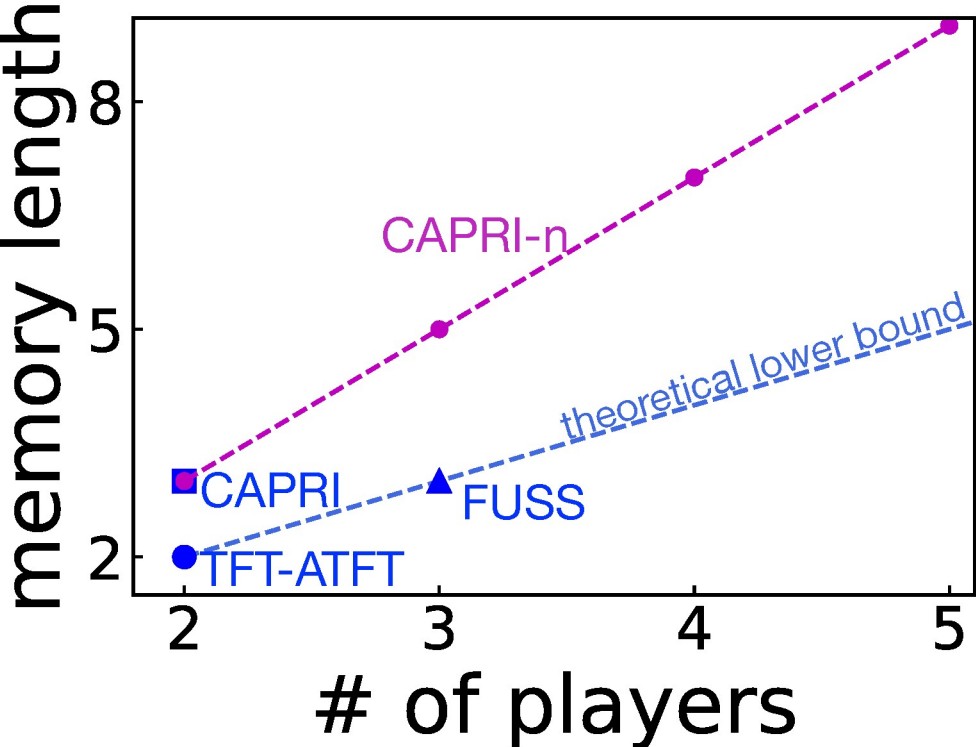

**Fig 1. Memory length *m* required for each of currently known friendly-rivalry strategies in the *n*-person PG game** [18, 19, 21]. The dashed blue line depicts a theoretical lower bound $m = n$ for friendly rivalry [19], and the strategy proposed in this work, called CAPRI-*n*, has $m = 2n - 1$.

## Materials and methods

In this section, we define the game and construct a friendly-rivalry strategy by reasoning. See S1 Table for a summary of mathematical symbols.

### Public-goods game

Let us consider the $n$-person public-goods (PG) game, in which a player may choose either cooperation ($c$), by contributing a token to a public pool, or defection ($d$), by refusing it. Let the number of cooperators be denoted as $n_c$. The $n_c$ tokens in the public pool are multiplied by a factor of $\rho$, where $1 < \rho < n$, and then equally redistributed to the $n$ players. We assume that the tokens are infinitely divisible. A player's payoff is thus given as

$$\begin{cases} \frac{\rho n_c}{n} & \text{when the player chooses } c, \\ 1 + \frac{\rho n_c}{n} & \text{when the player chooses } d. \end{cases} \tag{1}$$

Clearly, it is always better off to choose $d$ regardless of $n_c$, so full defection is the only Nash equilibrium of this one-shot game. In this study, this game will be repeated indefinitely with no discounting factor to facilitate direct reciprocity. Every player can choose an action between $c$ and $d$ by referring to the previous $m$ rounds. At the same time, a player can make implementation error, e.g., by choosing $d$ while intending $c$ and vice versa, with small probability $\epsilon \ll 1$.

### Axiomatic approach

The long-term payoff of player $X$ is defined as

$$\Pi_X \equiv \lim_{T \to \infty} \frac{1}{T} \sum_{t=0}^{T-1} \pi_X^{(t)}, \tag{2}$$

where $\pi_X^{(t)}$ is player $X$'s instantaneous payoff in round $t$. If $\epsilon > 0$, the Markovian dynamics of strategic interaction for a given strategy profile converges to a unique stationary distribution, from which $\Pi_X$ can readily be calculated [24, 25]. In terms of the players' long-term payoffs, we wish to propose the following three criteria that a successful strategy $\Omega$ should satisfy [18, 19, 21, 26].

1. Efficiency: Mutual cooperation must be achieved with probability one as $\epsilon \to 0$ if all the players have adopted $\Omega$. In other words, this criterion requires $\lim_{\epsilon \to 0^+} \Pi_X = \rho$ when the strategy profile $\mathcal{P} = \{\Omega, \Omega, \dots, \Omega\}$.

2. Defensibility: It must be guaranteed that none of the co-players can obtain higher long-term payoffs against $\Omega$ regardless of their strategies and initial states when $\epsilon = 0$. It implies that $\lim_{\epsilon \to 0^+} (\Pi_X - \Pi_Y) \geq 0$, where player $X$ is using strategy $\Omega$ and $Y$ denotes any possible co-player of $X$.

3. Distinguishability: If $X$ uses $\Omega$ and all the co-players are unconditional cooperators (AllC), player $X$ can exploit them to earn a strictly higher long-term payoff than theirs. That is, $\Pi_X > \Pi_Y$ when $Y$ is an AllC player.

When a strategy satisfies defensibility and efficiency, the strategy is a friendly rival. A symmetric strategy profile which consists of a friendly-rivalry strategy forms a cooperative Nash equilibrium [18, 19, 21], and the proof is straightforward: Assume that everyone initially uses a friendly-rivalry strategy $\Omega$ with earning $\rho$ per round. If one player, say, $X$, changes his or her strategy alone, $X$'s payoff will change to $\Pi_X$, while each of the co-players earns $\Pi_\Omega$.

Defensibility guarantees that $\Pi_X \leq \Pi_\Omega$, and full cooperation is Pareto-optimal, i.e., $(n-1)\Pi_\Omega + \Pi_X \leq n\rho$. Combining these two inequalities, we see that

$$(n-1)\Pi_X + \Pi_X \leq (n-1)\Pi_\Omega + \Pi_X \leq n\rho, \tag{3}$$

which means that $\Pi_X \leq \rho$. The player cannot increase his or her payoff by deviating from $\Omega$ alone. The third criterion is a requirement to suppress invasion of AllC due to neutral drift in the evolutionary context [27–29]. We call a strategy 'successful' if it meets all the above three criteria. Depending on the definition of successfulness, one could choose a different set of axioms for an alternative characterization [30].

## Strategy design

Let us construct a deterministic strategy with memory length $m = 2n - 1$ and show that the proposed strategy indeed meets all of the above three criteria. In the following, we will take an example of three players ($n = 3$) who are called Alice ($A$), Bob ($B$), and Charlie ($C$), respectively, and choose Alice as a focal player playing this strategy.

Before proceeding, it is convenient to introduce some notations for the sake of brevity. The three players' history profile over the previous $m = 5$ rounds can be represented as $h_t = (A_{t-5} A_{t-4} A_{t-3} A_{t-2} A_{t-1}; B_{t-5} B_{t-4} B_{t-3} B_{t-2} B_{t-1}; C_{t-5} C_{t-4} C_{t-3} C_{t-2} C_{t-1})$, where $A_\tau, B_\tau$, and $C_\tau$ denote their respective actions at round $\tau$. The last round of full cooperation will be denoted by $t^*$. According to the payoff definition [Eq (1)], we can fully determine Alice's cumulative payoff over a given period, $\sum_t \pi_A^{(t)}$, just by counting how many times each of the players has defected during the period. This is due to the linearity of operations acting on the number of tokens: The tokens contributed to the public pool are multiplied by a constant factor $\rho$ and equally distributed to all the players, and Alice saves a token every time she defects. For example, if all the players have defected the same number of times, their payoffs must be the same irrespective of the exact history. We thus introduce $\Delta_A^{\tau_1, \tau_2}$ to denote Alice's number of defections in $[\tau_1, \tau_2]$. Likewise, we can define $\Delta_B^{\tau_1, \tau_2}$ for Bob and $\Delta_C^{\tau_1, \tau_2}$ for Charlie. We also define $N_d$ as the maximum difference among the players in numbers of defections over the previous $m$ rounds:

$$N_d \equiv \max_{i,j \in \{A,B,C\}} \left| \Delta_i^{t-m,t-1} - \Delta_j^{t-m,t-1} \right|. \tag{4}$$

With these notations, we can now design a successful strategy satisfying all the three criteria simultaneously. To this end, we divide the set of history profiles into three mutually exclusive cases: The first case is that full cooperation occurred in the last round ($t^* = t - 1$). The second case is that it is not in the last round but still in their memory ($t - m \leq t^* < t - 1$). The third case is that no player remembers the last round of full cooperation ($t^* < t - m$). Let us consider these cases one by one, together with adequate rules for each.

1. $t^* = t - 1$

   - Cooperate: If this is the case, Alice has to choose $c$ under the condition that $N_d < n$. For example, the inequality is true for ($ccccc;cccdc;ccccc$), for which $N_d = 1$. On the other hand, it is not true for ($cdddc;ccddc;ccccc$) because its $N_d$ is equal to $n = 3$.

2. $t - m \leq t^* < t - 1$

   - Accept: Alice has to accept punishment from the co-players by choosing $c$, under the condition that $\Delta_A^{t^*,t-1} \geq \Delta_B^{t^*,t-1}$ and $\Delta_A^{t^*,t-1} \geq \Delta_C^{t^*,t-1}$ in addition to $N_d < n$. For example, $c$ will be prescribed to Alice at ($cccdc;ccccd, ccccc$), where we have $t^* = t - 3$, $\Delta_A^{t^*,t-1} = \Delta_B^{t^*,t-1} = 1$,

$\Delta_C^{t^*,t-1} = 0$, and $N_d = 1$, which satisfies the above inequalities. On the other hand, the condition is not met by (*ccddd*;*ccddd*;*ccccc*) which gives $N_d = 3$.

- Punish: Alice has to punish the co-players by choosing *d*, under the condition that $\Delta_A^{t^*,t-1} < \Delta_B^{t^*,t-1}$ or $\Delta_A^{t^*,t-1} < \Delta_C^{t^*,t-1}$ in addition to $N_d < n$. For example, *d* is prescribed at (*cccd*;*cccdd*;*ccccc*) because $N_d = 2$ and Alice has defected fewer times than Bob since the last round of full cooperation at $t^* = t - 3$.

3. $t^* < t - m$

- Recover: Alice has to recover cooperation by choosing *c*, under the condition that all the players except one cooperated in the last round. For $n = 3$, it means (*ddddd*;*ddddc*;*ddddc*) and its permutations.

4. In all the other cases, defect.

A strategy of this sort for the *n*-person PG game will be called CAPRI-*n* after the first letters of the five constitutive rules. Note that these five rules may be implemented in a number of different ways [21], and we take this way because it provides the most straightforward way to prove the three criteria. Each of the rules can also be regarded as the player's internal *state* consisting of multiple history profiles [26]. For example, Alice can find herself at state R, the abbreviation for 'Recover', when her history profile is (*ddddd*;*ddddc*;*ddddc*), at which she must choose *c*. The connection structure of the above five states is graphically represented in Fig 2, which is helpful for understanding how defensibility and efficiency are realized as shown below.

**Defensibility.** Let us begin by checking defensibility. Our CAPRI-*n* player Alice cooperates only at states C, A, and R, so the question is whether she can be forced to visit one of these states repeatedly with giving a strictly higher payoff to one of her co-players. If Alice's state is C, it means that everyone cooperated at $t - 1$. If some of her co-players defect from this full cooperation at *t*, she will retaliate at $t + 1$ with state P, so she suffers from unilateral defection at most once. Full cooperation is already broken, so it must be only through state A or R if she comes back to C. The former case, i.e. the case when she comes back to C via A, means that Alice has already been compensated for the payoff loss: Otherwise, she would not have state A. In the latter case when C is accessed via R, on the other hand, the only possible history profile is (*ddddd*;*ddddc*;*ddddc*) unless she made a mistake, which means the compensation has been done in the last round. Finally, state A can be accessed from states P and I, at both of which one cannot exploit Alice who chooses *d*. To sum up, it is impossible to see unilateral cooperation of a CAPRI-*n* player repeatedly.

**Efficiency.** The next criterion is efficiency. Provided that CAPRI-*n* is employed by all the players, only full cooperation or full defection can be a stationary state, and we can verify this statement by checking each possible case:

- If $t^* = t - 1$, everyone has to cooperate again as prescribed at state C, so full cooperation will continue.

- If $t - m \le t^* < t - 1$ and $N_d < n$, some players must be at state A while the others are at state P. The latter players at P will keep defecting until satisfying $\Delta_A^{t^*,t-1} = \Delta_B^{t^*,t-1} = \Delta_C^{t^*,t-1}$. If they make it with keeping $t^* \ge t - m$, all of them should choose *c* as prescribed at state A, and the resulting mutual cooperation will continue. If they don't, the situation to everyone reduces to state I, at which they will defect over and over.

- The remaining state is R, but it is always transient.

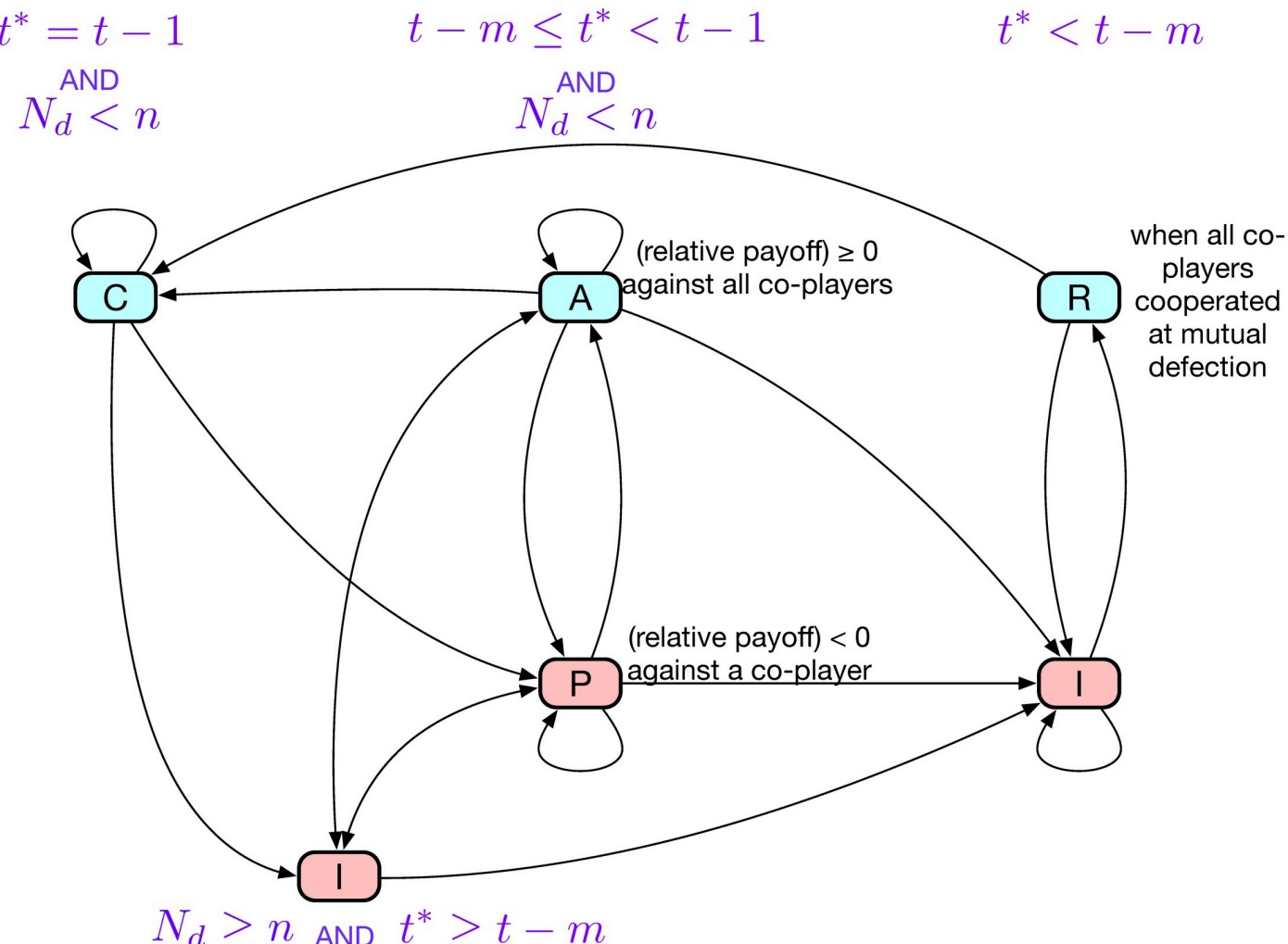

**Fig 2. Schematic diagram of the transition between states of CAPRI-*n*.** The five rules of the strategy can be identified with the player's internal states [26], each of which is represented as a node in this diagram. An exception is state I, which corresponds to two nodes to clarify the following point: When $t^* \geq t - m$, the state may have outgoing connections to A and P. When $t^* < t - m$, on the other hand, the only possible next state is R. The player has to choose *c* at a blue node and *d* at a red node. We have omitted error-caused transitions for the sake of simplicity.

To judge efficiency, we need to consider error-caused transition between these two stationary states, i.e., full cooperation and full defection. The transition from the latter to the former is possible only through state R, which occurs with probability of $O(\epsilon^{n-1})$. On the other hand, full cooperation can be made robust against every possible type of $(n - 1)$-bit error if $m = 2n - 1$. To see the basic idea, let us suppose that memory is just long enough, and it will immediately become clear what we mean by 'enough'. If an initial state of full cooperation is disturbed by error, everyone goes to either P (punish) or A (accept), and what happens is the following: Those who have defected more will accept punishment from the others, so the players' numbers of defections tend to be equalized as time goes by. When everyone has finally defected the same number of times, they all arrive at state A, where *c* is the prescribed action. Then, according to the first rule C, full cooperation will continue. If we look at our example with Alice, Bob, and Charlie ($n = 3$), we basically mean that our strategy corrects every single-bit or double-bit error when its memory length is given as $m = 2n - 1 = 5$: To see how this happens, let us consider some possible types of error case by case.

1. Single-bit error: Imagine that a player, say, Alice, mistakenly defects from full cooperation at $t = 1$. She will have state A at $t = 2$, while the others have state P, so their payoffs should be equalized at $t = 3$ as a result of punishment, by which mutual cooperation is recovered. This scenario can be represented as follows:

$$
\begin{array}{llllll}
& t = 1 & & t = 2 & & t = 3 \\
A: & ccc\underline{d} & \xrightarrow{A} & cccdc & \xrightarrow{A} & ccdcc \\
B: & ccccc & \xrightarrow{P} & ccccd & \xrightarrow{A} & cccdc \\
C: & ccccc & \xrightarrow{P} & ccccd & \xrightarrow{A} & cccdc,
\end{array}
\tag{5}
$$

where the underline means a mistaken action, and the letter below each arrow means which rule applies there. The important point is that a single-bit error is corrected only in two rounds.

2. Double-bit error: In this case, we have several possibilities. First, we consider two players' simultaneous mistakes, which are corrected in a similar way to Eq (5).

$$
\begin{array}{llllll}
& t = 1 & & t = 2 & & t = 3 \\
A: & ccc\underline{d} & \xrightarrow{A} & cccdc & \xrightarrow{A} & ccdcc \\
B: & ccc\underline{d} & \xrightarrow{A} & cccdc & \xrightarrow{A} & ccdcc \\
C: & ccccc & \xrightarrow{P} & ccccd & \xrightarrow{A} & cccdc.
\end{array}
\tag{6}
$$

As another possibility, let us assume that Alice defects by mistake for two successive rounds. It is a simple extension of the recovery pattern in Eq (5):

$$
\begin{array}{llllllll}
& t = 1 & & t = 2 & & t = 3 & & t = 4 \\
A: & ccc\underline{d} & \xrightarrow{A} & ccc\underline{d}d & \xrightarrow{A} & ccddc & \xrightarrow{A} & cddcc \\
B: & ccccc & \xrightarrow{P} & ccccd & \xrightarrow{P} & cccdd & \xrightarrow{A} & ccddc \\
C: & ccccc & \xrightarrow{P} & ccccd & \xrightarrow{P} & cccdd & \xrightarrow{A} & ccddc.
\end{array}
\tag{7}
$$

It makes little difference whether error occurs to a single player twice in a row or it does to one after another:

$$
\begin{array}{llllllll}
& t = 1 & & t = 2 & & t = 3 & & t = 4 \\
A: & ccc\underline{d} & \xrightarrow{A} & cccdc & \xrightarrow{A} & ccdcc & \xrightarrow{A} & cdccc \\
B: & ccccc & \xrightarrow{P} & cccc\underline{c} & \xrightarrow{P} & ccccd & \xrightarrow{A} & cccdc \\
C: & ccccc & \xrightarrow{P} & ccccd & \xrightarrow{A} & cccdc & \xrightarrow{A} & ccdcc.
\end{array}
\tag{8}
$$

The last possibility to consider is when error occurs again at the end of Eq (5):

$$
\begin{array}{llllllllll}
& t = 1 & & t = 2 & & t = 3 & & t = 4 & & t = 5 \\
A: & ccc\underline{d} & \xrightarrow{A} & cccdc & \xrightarrow{A} & ccdc\underline{d} & \xrightarrow{A} & cdcdc & \xrightarrow{A} & dcdcc \\
B: & ccccc & \xrightarrow{P} & ccccd & \xrightarrow{A} & cccdc & \xrightarrow{P} & ccdcd & \xrightarrow{A} & cdccc \\
C: & ccccc & \xrightarrow{P} & ccccd & \xrightarrow{A} & cccdc & \xrightarrow{P} & ccdcd & \xrightarrow{A} & cdccc,
\end{array}
\tag{9}
$$

which needs additional two rounds to reach full cooperation at $t = 5$. Among all types of double-bit error, the last pattern and the like (i.e., error occurs again when cooperation is

about to be recovered) are the ones that require the longest memory for full recovery: If the distance between two errors is longer than two rounds, they can be regarded as two single-bit errors, which are separately correctable [Eq (5)]. In general, if we have to correct ($n-1$)-bit error that occurs every other round, the memory length $m = 2(n-1) + 1$ is required in total, where the last bit has been added to memorize the last round of full cooperation. It is also enough to correct simpler types of error as illustrated above. To sum up, with $m = 2n - 1$, the transition probability from mutual cooperation to defection can be suppressed down to $O(\epsilon^n)$, whereas the transition in the opposite direction through R has probability of $O(\epsilon^{n-1})$. Therefore, the players form full cooperation in the limit of $\epsilon \to 0$, fulfilling the efficiency criterion.

**Distinguishability.**   The last criterion is distinguishability. If the others are AllC players, our CAPRI-*n* player will continue unilateral defection when she defected $n$ consecutive times by error, as prescribed by rule I. One can escape from such a state with probability of $O(\epsilon^n)$ due to the condition of $N_d < n$ in rule C, so this stationary state coexists with full cooperation in the limit of $\epsilon \to 0$.

## Evolutionary simulation

We consider a standard stochastic model proposed in [29], where a well-mixed population of size $N$ evolves over time by an imitation process. A key assumption of this model is that the mutation rate is low so that at most one mutant strategy can exist in the resident population. In other words, the time that it takes to go extinct or occupy the whole population by selection is assumed to be much shorter than the time scale of mutation. Let us assume that a mutant strategy $x$ is introduced to a population of strategy $y$. The population dynamics is modeled by the frequency-dependent Moran process, in which the fixation probability of the mutant is given in a closed from:

$$\phi_{xy} = \left( \sum_{i=0}^{N-1} \prod_{j=1}^{i} \Gamma_j \right)^{-1} \tag{10}$$

with $\Gamma_j \equiv P_{j,j-1}/P_{j,j+1}$, where $P_{j,j\pm1}$ denotes the probability that the number of mutants increases or decreases from $j$ by one.

For $n = 2$, the fixation probability is calculated in the following way: Suppose that we have $j$ individuals of the mutant strategy and $N - j$ individuals of the resident strategy. If we randomly choose a mutant and a resident individual, their average payoffs are obtained as

$$\begin{cases} s_x = \frac{1}{N-1}\left[ (j-1)s_{xx} + (N-j)s_{xy} \right] \\ s_y = \frac{1}{N-1}\left[ (N-j-1)s_{yy} + js_{yx} \right], \end{cases} \tag{11}$$

respectively, where $s_{\alpha\beta}$ is $\alpha$'s long-term payoff against $\beta$. According to the imitation process, $x$ can change to $y$ with probability $f_{x \to y}$ defined as follows:

$$f_{x \to y} = \frac{1}{1 + \exp\left[\sigma(s_x - s_y)\right]}, \tag{12}$$

where $\sigma$ means the strength of selection. Then, we have

$$\Gamma_j = \exp\left[\sigma(s_y - s_x)\right], \tag{13}$$

and the fixation probability is calculated as

$$\phi_{xy}^{-1} \quad = \quad \sum_{i=0}^{N-1}\prod_{j=1}^{i}e^{\sigma[(N-j-1)s_{yy}+js_{yx}-(j-1)s_{xx}-(N-j)s_{xy}]/(N-1)} \tag{14}$$

$$= \quad \sum_{i=0}^{N-1}e^{\sigma i[(-i+2N-3)s_{yy}+(i+1)s_{yx}-(-i+2N-1)s_{xy}-(i-1)s_{xx}]/[2(N-1)]}. \tag{15}$$

If $y$ is a friendly rival, i.e. if $s_{yy} \geq s_{xx}$ and $s_{yy} \geq s_{xy}$ in addition to $s_{yx} \geq s_{xy}$, Jensen's inequality shows that $\phi_{xy} \leq 1/N$ for arbitrary $x$, indicating that $y$ has evolutionary robustness for any $N$, $\rho$, and $\sigma$ [21].

For $n = 3$, the fixation probability is calculated in a similar way. We randomly pick up three players from a well-mixed population, and the respective average payoffs of playing $x$ and $y$ can be written by using the binomial coefficients as follows:

$$\begin{cases} s_x = \frac{1}{(N-1)(N-2)}\left[\binom{j-1}{2}s_{xxx} + \binom{j-1}{1}\binom{N-j}{1}s_{xxy} + \binom{N-j}{2}s_{xyy}\right] \\ s_y = \frac{1}{(N-1)(N-2)}\left[\binom{j}{2}s_{yxx} + \binom{N-j-1}{1}\binom{j}{1}s_{yyx} + \binom{N-j-1}{2}s_{yyy}\right], \end{cases} \tag{16}$$

where $s_{\alpha\beta\gamma}$ is $\alpha$'s long-term payoff against $\beta$ and $\gamma$. Plugging these expressions into Eqs (10) and (13), one can calculate the fixation probability $\phi_{xy}$ for the three-person case as well. Differently from the two-person case, however, friendly rivalry itself does not necessarily guarantee evolutionary robustness if $n \geq 3$: Assume that a friendly-rivalry strategy $y$ cannot distinguish a mutant $x$, whereas $x$ does distinguish $y$ when $x$ forms the majority of the $n$-person game. If $n = 3$, for example, it means that $s_{yyx} = s_{xyy} = \rho$ whereas $s_{yxx} = s_{xxy} = 0$. Furthermore, if the mutants are efficient among themselves, i.e., $s_{xxx} = \rho$, then its fixation probability will be higher than $1/N$. As of now, we find no reason to rule out the possibility of such a mutant.

We can interpret $\phi_{xy}$ as transition probability from $y$ to $x$ from the viewpoint of the population. From the stationary distribution of this Markovian dynamics, we can thus calculate abundance of each available strategy in a numerically exact manner [31, 32]. For the sake of simplicity, we use the donation game as a simplified form of the PD game as well as its generalization to $n$ players in the numerical calculation. That is, with the benefit of cooperation $b > 1$, each player can donate $b/(n - 1)$ to each co-player at the unit cost, which corresponds to $\rho = nb/[b + (n - 1)]$ up to scaling. In the next section, we will present numerical results obtained by using OACIS [33].

## Results

### Friendly rivalry

To check the validity of our construction, we computationally examined the three criteria by using graph-theoretic calculations used in [19, 21, 34]. For $n = 2$, we directly confirmed that CAPRI-2 is indeed a successful strategy satisfying all the three criteria. For $n = 3$, we conducted mapping to an automaton to obtain a simplified yet equivalent graph representation [26] to reduce computational complexity, and our graph-theoretic calculation confirmed that the resulting automaton indeed passed all the criteria. For $n = 4$, the required amount of calculation to directly check the criteria was beyond our computational resources, so we employed a Monte Carlo method to simulate the game. The Monte Carlo method was also used to double-check the performance of CAPRI-2 and CAPRI-3. See S1 Appendix for more discussion on computational details.

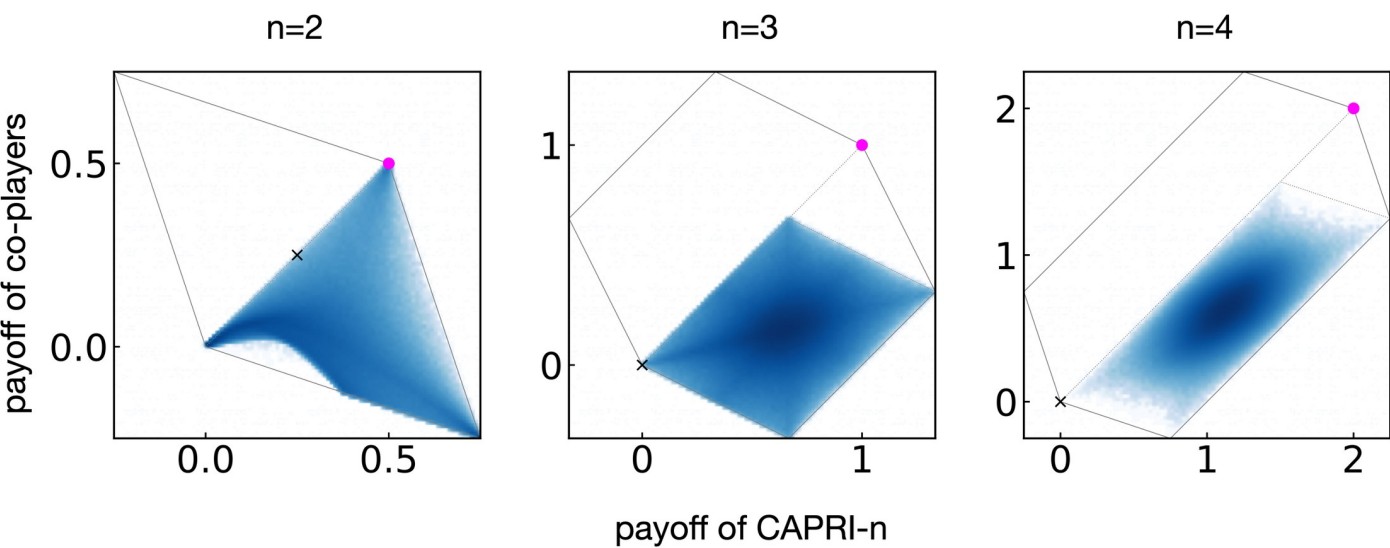

**Fig 3. Distribution of long-term payoffs when a CAPRI-*n* player meets co-players whose $p_{\mu\nu}$'s are randomly sampled from the unit interval.** Darker shades toward blue indicate higher frequency of occurrence. The multiplication factors for *n* = 2, 3, and 4 are 1.5, 2, and 3, respectively, and the solid lines indicate the region of feasible payoffs. In each case, the filled circle means the long-term payoffs when CAPRI-*n* is adopted by all the players, whereas the cross shows those of TFT players as a reference point. In each panel, we have drawn a dotted line along the diagonal as a simple check for defensibility. For *n* = 3 or 4, the parallelogram surrounding the blue area indicates the set of feasible payoffs when the focal player is AllD, which indicates that the behavior of CAPRI-*n* is similar to AllD against most of the memory-one players.

The Monte Carlo calculation was performed as follows: Let us denote a memory-one strategy as $(p_{cc}, p_{cd}, p_{dc}, p_{dd})$ where $p_{\mu\nu}$ means the player's probability to cooperate when the player and the co-player did $\mu$ and $\nu$, respectively, in the previous round. The initial $\mu$ and $\nu$ can be omitted in the strategy description because they are irrelevant to the long-term payoff as long as $\epsilon > 0$. Fig 3 shows the distribution of payoffs when Alice used CAPRI-*n* whereas each of her co-players' strategies was composed of four $p_{\mu\nu}$'s randomly sampled from the unit interval. The co-players' payoffs never exceeded Alice's, as required by defensibility.

We also calculated the probability of full cooperation for *n* = 2, 3 and 4 when CAPRI-*n* was adopted by all the players in order to check efficiency. By using linear-algebraic [18, 19] or Monte Carlo calculation, with $\epsilon = 10^{-4}$, we obtained 0.999, 0.997, 0.978 for *n* = 2, 3, and 4, respectively, which supports the conclusion that they all satisfy the efficiency criterion.

## Evolutionary performance

Before checking the evolutionary performance of our proposed strategy, we conducted simulations without CAPRI-*n* for comparison. Figs 4A and 5A show the results when the strategies were sampled from deterministic memory-one for *n* = 2 and 3. When *b* was low and/or *N* was small, defensible strategies such as AllD tended to be favored by selection, and the resulting cooperation level was low. On the other hand, when *b* or *N* was large, efficient strategies were favored, and they achieved a high level of cooperation. The reason is that cooperative strategies maintained high payoffs by interacting with many other cooperators even if they were exploited by a small number of aggressive mutants.

When CAPRI-*n* was introduced, it occupied a large amount of the population as shown in Figs 4B and 5B. Whereas each memory-one strategy flourished depending on the environmental parameters *b* and *N*, CAPRI-*n* was found abundant in the entire parameter region. In particular, it is striking that CAPRI-3 overwhelms all the other strategies in the three-person PG game for any moderate sizes of *b* and *N* (Fig 5B).

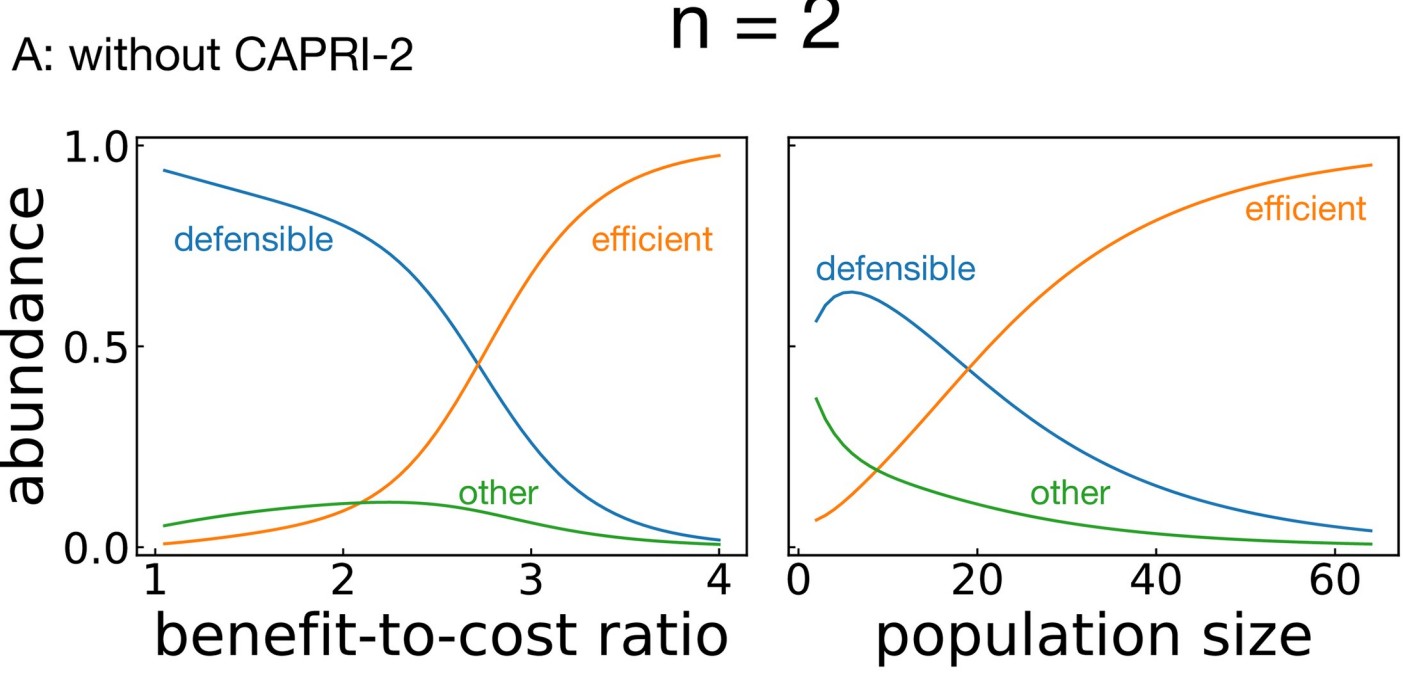

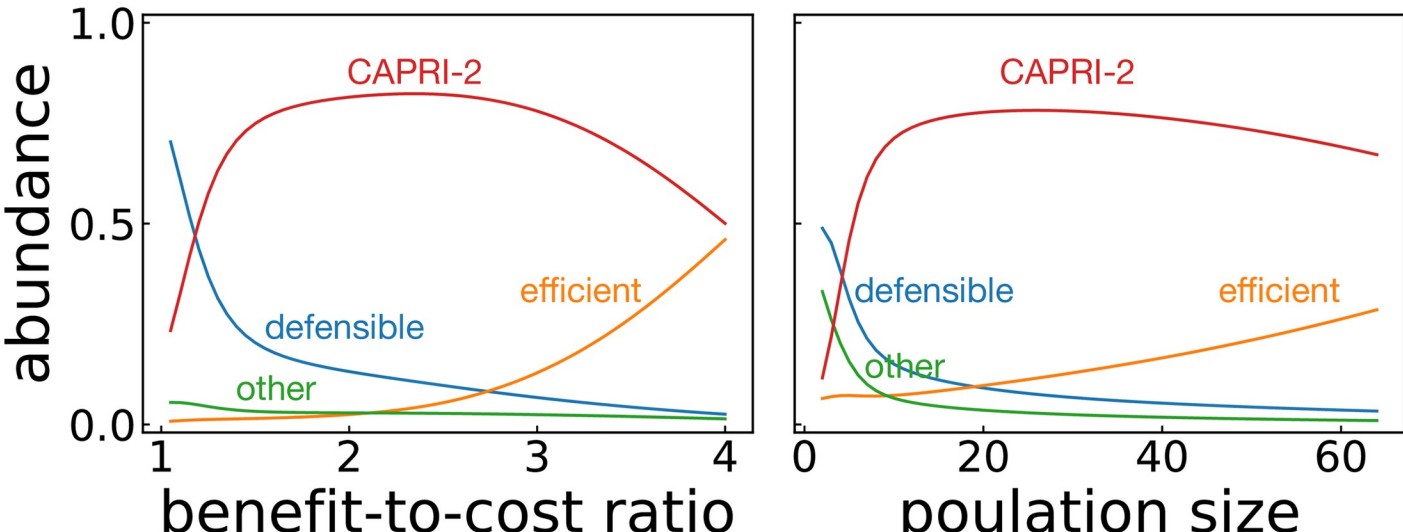

**Fig 4. Abundance of strategies for *n* = 2 as the benefit-to-cost ratio *b* and the population size *N* vary.** The default values were *b* = 3 and *N* = 30 unless otherwise specified. The strength of selection and the error probability were set to be $\sigma$ = 1 and $\epsilon = 10^{-4}$, respectively. (A) Simulation result with 16 memory-one deterministic strategies, classified into three categories, i.e., efficient, defensible, and the other strategies. (B) Effect of CAPRI-2 when it was added to the available set of strategies.

It is nevertheless worth pointing out that CAPRI-2 gave more and more room to efficient strategies in the iterated PD game as *b* or *N* increases (Fig 4B), and this is due to neutral drift: Although CAPRI-2 earns a strictly higher long-term payoff than AllC = (1, 1, 1, 1) and Win-Stay-Lose-Shift (WSLS) = (1, 0, 0, 1), it does not with respect to (1, 1, 1, 0), which can, in turn,

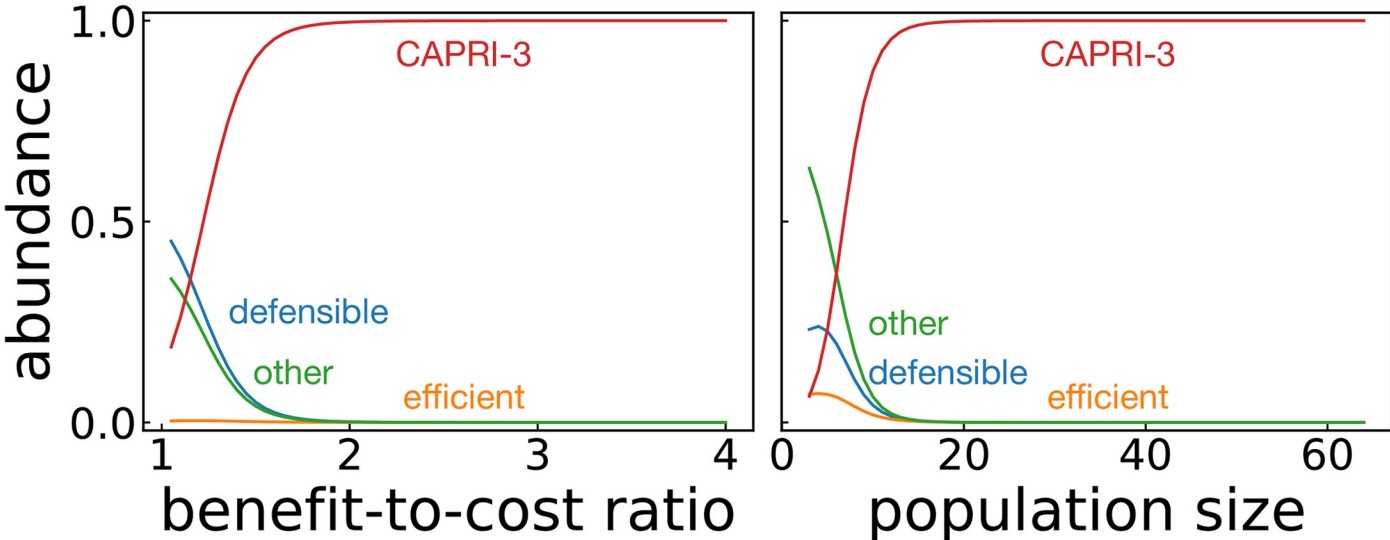

**Fig 5. Abundance of strategies for *n* = 3 as the benefit-to-cost ratio *b* and the population size *N* vary.** The default values were *b* = 3 and *N* = 30 unless otherwise specified. The strength of selection and the error probability were set to be $\sigma = 1$ and $\epsilon = 10^{-4}$, respectively. (A) Simulation result with 64 memory-one deterministic strategies, classified into three categories, i.e., efficient, defensible, and the other strategies. (B) Effect of CAPRI-3 when it was added to the available set of strategies.

be invaded by WSLS. For this reason, WSLS can become abundant in the presence of (1, 1, 1, 0) when the environmental conditions are favorable.

We also tested the performance of CAPRI-*n* against strategies with the same memory length. An obvious problem is the huge number of possible strategies: Provided that $m = 2n - 1$, the

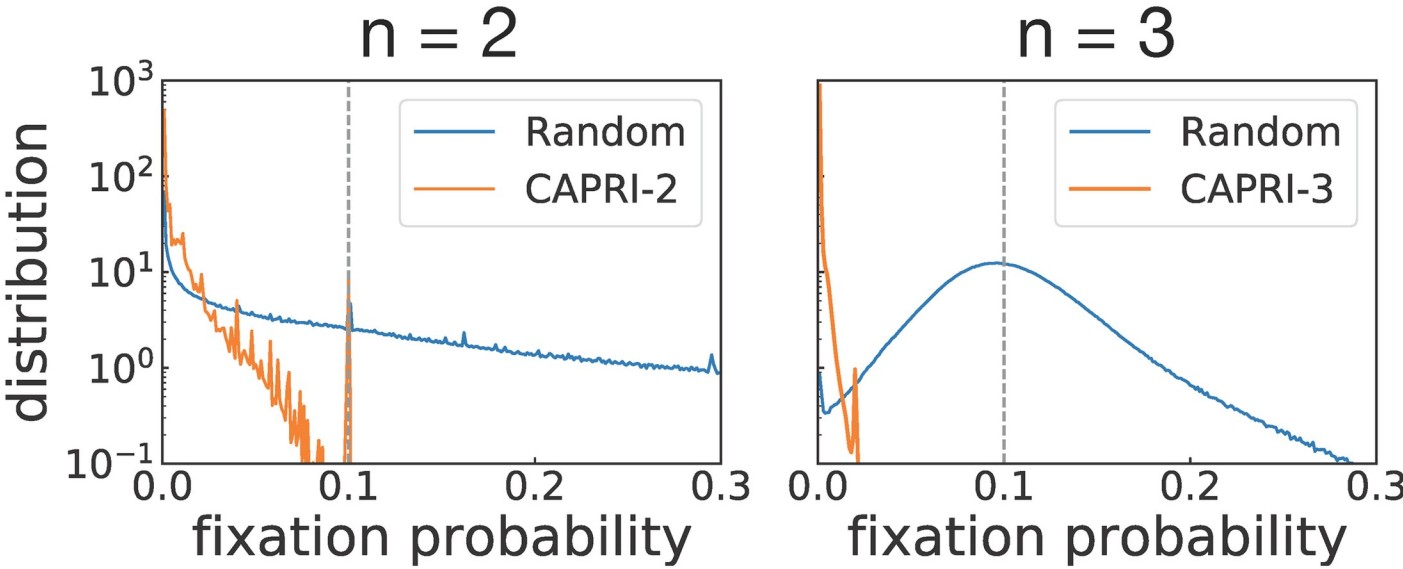

**Fig 6. Normalized distribution of fixation probabilities of mutants, which were randomly sampled from the set of deterministic strategies with the same memory length as CAPRI-*n*'s.** When we simulated the two-person game with taking CAPRI-2 as the resident strategy, $10^9$ mutants were sampled. In case of the three-person game in which CAPRI-3 was the resident strategy, the number of sampled mutants was $5 \times 10^6$. In either case, no mutant had higher fixation probability than $1/N$ (the vertical dashed line). On the other hand, when the resident was randomly drawn from the same strategy set, mutants frequently achieved fixation with probability higher than $1/N$. For each random sample of the resident strategy, $10^2$ mutants were tested, and this process was repeated $10^7$ and $10^5$ times when $n = 2$ and 3, respectively. Throughout this calculation, we used $N = 10$ as the population size, $\epsilon = 10^{-4}$ as the error probability, $\sigma = 1$ as the selection strength, and $b = 2$ as the benefit of cooperation.

number amounts to $2^{2^{nm}} \sim 10^{19}$ for $n = 2$, which grows to $10^{9864}$ for $n = 3$. As an alternative to exhaustive enumeration, we calculated fixation probabilities of mutants that are randomly sampled from the set of deterministic strategies with $m = 2n - 1$, with taking CAPRI-*n* as the resident strategy. The numbers of sampled mutants were $10^9$ and $5 \times 10^6$ for $n = 2$ and 3, respectively. As shown in Fig 6, none of them had fixation probability greater than $1/N$, and the tendency was more pronounced in the three-person game than in the two-person case. For comparison, we also tested resident strategies drawn randomly from the same strategy set with $m = 2n - 1$, and a significant fraction of mutants succeeded in fixation with probability higher than $1/N$. Although we have no proof for evolutionary robustness for $n \geq 3$, the numerical result suggested that it would be extremely unlikely to see CAPRI-*n* invaded by random mutants even if they had the same memory length.

## Discussion

In summary, we have constructed a friendly-rivalry strategy for the iterated *n*-person PG game. It maintains a cooperative Nash equilibrium in the presence of implementation error with probability $\epsilon \ll 1$, and it shows excellent evolutionary performance regardless of the environmental conditions such as the population size and the strength of selection. In this sense, the *n*-person social dilemma is solved. The strategy requires memory of the previous $m = 2n - 1$ rounds and consists of the following five rules: Cooperate if everyone did, accept punishment for your own mistake, punish others' defection, recover cooperation if you find a chance, and defect in all the other circumstances.

Although we have considered only implementation error, perception error can also be corrected if it occurs with sufficiently low probability: The disagreement between the players' history profiles due to the perception error will soon be removed at full defection, and the players

will escape from mutual defection with probability of $O(\epsilon^n)$. Unless another perception error perturbs this process, the players will eventually arrive at full cooperation, overcoming the perception error.

Another important solution concept to the *n*-person dilemma can be derived from a different set of criteria: By requiring mutual cooperation, error correction, and retaliation with a time scale of *k* rounds, one can characterize the all-or-none (AON-*k*) strategy, which is defined as prescribing *c* only when everyone cooperated or no one did in each of the previous *k* rounds [30, 35, 36]. For example, WSLS = (1, 0, 0, 1) is equivalent to AON-1. For each *k*, one can find a threshold of the multiplication factor above which AON-*k* constitutes a subgame-perfect equilibrium [30]. AON-*k* performs well in evolutionary simulation because it prescribes *d* as the default action, just as CAPRI-*n* does in state I, unless the players have synchronized their behavior over the previous *k* rounds. As a result, it earns a strictly higher payoff against a broad range of strategies.

In general, CAPRI-*n* with $m = 2n - 1$ can repeatedly exploit the other co-players playing AON-*k* if $k < m - 1$, which means that an AON-*k* population can readily be invaded by CAPRI-*n* unless *k* is large enough. Considering the condition for AON-*k* to be subgame perfect, one could speculate that AON with small *k* can be abundant in an environment with a high multiplication factor. However, our finding implies that such a simple solution may not be sustained when CAPRI-*n* is available. This is especially crucial when population size is not large enough because AON-*k* lacks defensibility. Still, AON-*k* remains as a strong competitor to CAPRI-*n* in evolutionary simulation: For example, although WSLS earns a strictly less payoff against CAPRI-2, it circumvented the difficulty of fixation with the aid of a third strategy (1, 1, 1, 0).

We have assumed the small-$\epsilon$ limit, but an important question is how the performance of CAPRI-*n* will be affected if $\epsilon$ takes a finite value. One possibility is that it could set a limit on the maximum number of players in regard to the efficiency criterion: The transition probability from full defection to cooperation is given as $n\epsilon^{n-1}$ by construction, where the prefactor *n* originates from the number of possible ways to choose $(n - 1)$ players who will cooperate by mistake. The probability of transition in the opposite direction is of $O(\epsilon^n)$ at most, but it is reasonable to guess that it also has a prefactor as an increasing function of *n*. If it can be approximated as $n^\tau \epsilon^n$ with $\tau > 1$, for example, efficiency criterion should require $n \lesssim (1/\epsilon)^{1/(\tau-1)}$. To achieve cooperation among a large number of players with finite error probability, therefore, we may have to revise the rules so as to adjust the prefactors.

From a practical point of view, it is worth noting that the five rules of CAPRI-*n* mostly refer to two factors: One is the players' last action at $t - 1$, and the other is the differences in the players' respective numbers of defections over the previous *m* rounds. In other words, exact details of the history profile are irrelevant, and this point greatly reduces the cognitive burden to play this strategy. In fact, according to a recent experiment, people assign reputation to their co-players based mainly on their last action and their average numbers of defection [37]. This could explain the reason that such a delicate relationship called friendly rivalry can develop spontaneously and unwittingly among a group of people. How to keep such a relation healthy and productive has so far been acquired as tacit knowledge surrounded by anecdotes and experiences, and CAPRI-*n* expresses its essential how-tos in a form of explicit knowledge which can be designed, analyzed, and transmitted systematically.

## Supporting information

**S1 Appendix. Computational check for efficiency, defensibility, and distinguishability.**
(PDF)

**S1 Table. Summary of mathematical symbols used in this work.**
(PDF)

## Acknowledgments

Part of the results is obtained by using the Fugaku computer at RIKEN Center for Computational Science (Proposal number ra000002). We appreciate the APCTP for its hospitality during completion of this work.

## Author Contributions

**Conceptualization:** Yohsuke Murase.

**Investigation:** Yohsuke Murase.

**Methodology:** Yohsuke Murase, Seung Ki Baek.

**Software:** Yohsuke Murase.

**Validation:** Seung Ki Baek.

**Visualization:** Yohsuke Murase.

**Writing – original draft:** Yohsuke Murase.

**Writing – review & editing:** Seung Ki Baek.

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
