## [Decision Letter · Decision Letter 0]

3 Oct 2020

Dear Prof. Baek,

Thank you very much for submitting your manuscript "Friendly-rivalry solution to the iterated n-person public-goods game" for consideration at PLOS Computational Biology.

As with all papers reviewed by the journal, your manuscript was reviewed by members of the editorial board and by several independent reviewers. In light of the reviews (below this email), we would like to invite the resubmission of a significantly-revised version that takes into account the reviewers' comments.

The paper has now been revised by three of our reviewers, who have made a number of criticisms that are sufficiently adverse as to suggest that a major revision is due before a final decision about publication can be made. We therefore ask you to address all comments by the reviewers. However, we ask you to pay particular attention to:

- the technical aspects raised by reviewer 3. This might require to show that the results are robust beyond what is now presented in the MS.

- We would also like that similarities and differences with respect to the paper arXiv:2004.00261 were clearly stated. PCB only published highly original contributions, and we appreciate that you have studied the same strategy for the PD game, which apparently leads to very similar findings and even the discussion presented.

We cannot make any decision about publication until we have seen the revised manuscript and your response to the reviewers' comments. Your revised manuscript is also likely to be sent to reviewers for further evaluation.

Sincerely,

Yamir Moreno

Guest Editor

PLOS Computational Biology

Stefano Allesina

Deputy Editor

PLOS Computational Biology

The paper has now been revised by three of our reviewers, who have made a number of criticisms that are sufficiently adverse as to suggest that a major revision is due before a final decision about publication can be made. We therefore ask you to address all comments by the reviewers. However, we ask you to pay particular attention to:

- the technical aspects raised by reviewer 3. This might require to show that the results are robust beyond what is now presented in the MS.

- We would also like that similarities and differences with respect to the paper arXiv:2004.00261 were clearly stated. PCB only published highly original contributions, and we appreciate that you have studied the same strategy for the PD game, which apparently leads to very similar findings and even the discussion presented.

Reviewer's Responses to Questions

**Comments to the Authors:**

Reviewer #1: I found the study "Friendly-rivalry solution to the iterated n-person public-goods game" by Murase and Baek very interesting, if not important. I believe that the study should be published subject to addressing the issues listed below.

NON-NEGOTIABLE

• There are many mathematical symbols. A table of symbols is absolutely needed.

MAJOR

L209-225... I really struggled trying to understand the argument about efficiency. If I had more time, maybe I would fully grasp it, but as it is, the available information did not really convince me. I strongly suggest that the authors consider rephrasing the argument for the benefit of their readers.

L265-272... This whole paragraph sounds as if the authors were saying: "We checked our claims and you should trust us!" It is utterly unclear whether the checks were performed on paper or computationally. If the former is the case, then the authors should detail their calculations in an appendix or supporting text. If the latter is the case, then a reference to the precise part of the code should be made available.

Fig. 3... What do gray dots represent? What does blueish color coding stand for?

MINOR

L139... Eq. (1) does not really define a matrix.

L191, 192... I'm not sure if I got 'former' and 'latter' right at this particular spot in the manuscript. Perhaps the authors should rephrase.

L266... CAPRI-2 rather than CAPRI-n.

L366... The authors should cite OACIS in the Methods section.

Reviewer #2: This manuscript presents an interesting study on the social dilemma of cooperation by an approach based on the iterated n-person public-goods Game.

The manuscript is well presented. I found correct mathematics as well as the results obtained both via simulations and formal reasoning. However, I have some concerns about both the justification of the model and the discussion.

The authors extend the concept of Nash equilibrium to long-term payoffs. This extension has been adopted in recent literature, for example, as authors cited, in the context of zero-determinant strategies.

The authors define a set of conditions for a particular collaborative strategy (CAPRI). If I understand the model correctly, the condition for the Nash equilibrium, for the particular case of a symmetric strategic profile where all the agents share the strategy $\\Omega$, is reformulated as:

"It must be guaranteed that none of the co-players can obtain higher long-term payoffs against $\\Omega$ regardless of their strategies and initial states when e = 0."

Furthermore, the required memory length of players adopting such a strategy is m = 2n − 1. I find this concept far from the original Nash Equilibrium, and, at least, would expect a deep discussion by the authors on this (or perhaps to rename it). Nevertheless, the authors study the system for small values of n, which, one the one hand, make the memory requirement realistic while, on the other hand, bring this approach very close to that of pairwise interactions which has been addressed by the authors in Ref. 22.

The memory considered (2n − 1) is very long for large sets of co-players, furthermore when the authors state "not far from human behavior." Does it mean that, in an n-agents interaction, an agent may have a memory of 2n-1 previous interactions of everybody? I think that the authors should contextualize their model and justify it according to its particular requirements on memory length to differentiate it from that of pairwise interactions.

Reviewer #3: See attached PDF

**Have all data underlying the figures and results presented in the manuscript been provided?**

Reviewer #1: Yes

Reviewer #2: Yes

Reviewer #3: Yes

PLOS authors have the option to publish the peer review history of their article (what does this mean?). If published, this will include your full peer review and any attached files.

Reviewer #1: **Yes: **Marko Jusup

Reviewer #2: No

Reviewer #3: No
---

## [Decision Letter · Decision Letter 1]

12 Dec 2020

Dear Prof. Baek,

We are pleased to inform you that your manuscript 'Friendly-rivalry solution to the iterated n-person public-goods game' has been provisionally accepted for publication in PLOS Computational Biology.

Best regards,

Yamir Moreno

Guest Editor

PLOS Computational Biology

Stefano Allesina

Deputy Editor

PLOS Computational Biology

Reviewer's Responses to Questions

**Comments to the Authors:**

Reviewer #1: I'm satisfied with the authors' answers to my comments.

Reviewer #2: I appreciate the efforts made by the authors to improve the manuscript as well as the answers to the reviewers' suggestions.

Reviewers' concerns have revealed some weaknesses in the model. In my opinion, in addition to the improvements made in the current version and the necessary corrections, the authors have contextualized these weak points by making the limitations explicit.

The revised version of this paper has improved significantly the original and, in my opinion, it can be accepted for publication.

Reviewer #3: I thank the authors for their careful response to the queries I raised. Although I would be interested to see a further analysis of the role of errors in determining the evolutionary dynamics of the strategies presented in this paper, I agree that the changes made in #11 and #12 are sufficient to address the issues I raised, and I'm therefore happy to recommend accepting the paper

**Have all data underlying the figures and results presented in the manuscript been provided?**

Reviewer #1: Yes

Reviewer #2: Yes

Reviewer #3: Yes

PLOS authors have the option to publish the peer review history of their article (what does this mean?). If published, this will include your full peer review and any attached files.

Reviewer #1: No

Reviewer #2: No

Reviewer #3: **Yes: **Alexander J. Stewart

---

## [Editor Report · Acceptance letter]

18 Jan 2021

PCOMPBIOL-D-20-01362R1 

Friendly-rivalry solution to the iterated n-person public-goods game

Dear Dr Baek,

I am pleased to inform you that your manuscript has been formally accepted for publication in PLOS Computational Biology. Your manuscript is now with our production department and you will be notified of the publication date in due course.

With kind regards,

Jutka Oroszlan
